# INITIALIZING MODELS WITH LARGER ONES

**Zhiqiu Xu**[1], **Yanjie Chen**[2], **Kirill Vishniakov**[3], **Yida Yin**[2], **Zhiqiang Shen**[3],
**Trevor Darrell**[2], **Lingjie Liu**[1], **Zhuang Liu**[4]
[1]University of Pennsylvania  [2]UC Berkeley  [3]MBZUAI  [4]Meta AI Research

## ABSTRACT

Weight initialization plays an important role in neural network training. Widely used initialization methods are proposed and evaluated for networks that are trained from scratch. However, the growing number of pretrained models now offers new opportunities for tackling this classical problem of weight initialization. In this work, we introduce weight selection, a method for initializing smaller models by selecting a subset of weights from a pretrained larger model. This enables the transfer of knowledge from pretrained weights to smaller models. Our experiments demonstrate that weight selection can significantly enhance the performance of small models and reduce their training time. Notably, it can also be used together with knowledge distillation. Weight selection offers a new approach to leverage the power of pretrained models in resource-constrained settings, and we hope it can be a useful tool for training small models in the large-model era. Code is available at https://github.com/OscarXZQ/weight-selection.

## 1 INTRODUCTION

The initialization of neural network weights is crucial for their optimization. Proper initialization aids in model convergence and prevents issues like gradient vanishing. Two prominent initialization techniques, Xavier initialization (Glorot & Bengio, 2010) and Kaiming initialization (He et al., 2015), have played substantial roles in neural network training. They remain the default methods in modern deep learning libraries like PyTorch (Paszke et al., 2019).

These methods were developed for training neural networks from random initialization. At that time, it was the common practice. However, the landscape has changed. A variety of pretrained models are now readily available, thanks to collective efforts from the community (Wolf et al., 2019; Wightman, 2019). These models are trained on large datasets like ImageNet-21K (Deng et al., 2009) and LAION-5B (Schuhmann et al., 2022) and are often optimized by experts. As a result, fine-tuning from these pretrained models (Kolesnikov et al., 2020; Hu et al., 2022) is usually considered a preferred option today, rather than training models from scratch.

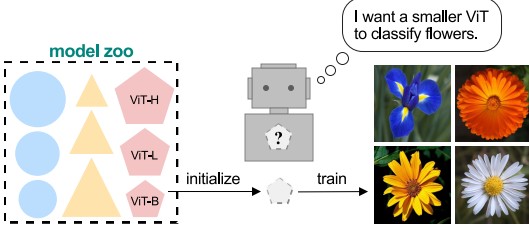

Figure 1: Large pretrained models offer new opportunities for initializing small models.

However, these pretrained large models can be prohibitive in their resource demand, preventing their wide adoption for resource-constrained settings, e.g., on mobile devices. For many pretrained model families, even the smallest model instance can be considered extremely large in certain contexts. For example, masked autoencoders (MAE) (He et al., 2022) and CLIP (Radford et al., 2021) both provide ViT-Base (Dosovitskiy et al., 2021), a 80M-parameter architecture, as their smallest pretrained Transformer model. This is already too large for applications on edge devices, and the smallest LLaMA (Touvron et al., 2023) model is even another 100 times larger, with 7B parameters. With few small pretrained models available, developers would have to train them from scratch on target datasets to suit their needs. This approach misses the opportunity to utilize large pretrained models, whose knowledge is learned from extensive training on large data.

In this work, we tackle this issue by introducing a weight initialization method that uses large pretrained models to train small models. Specifically, we introduce *weight selection*, where a subset of weights from a pretrained large model is selected to initialize a smaller model. This allows for knowledge learned by the large model to transfer to the small model through its weights. Thanks to the modular design of modern neural networks, weight selection involves only three simple steps: layer selection, component mapping, and element selection. This method can be applied to any smaller model within the same model family as the large model. Using weight selection for initializing a small model is straightforward and adds no extra computational cost compared to training from scratch. It could also be useful even for large model training, e.g., initializing a LLaMA-7B with trained weights from LLaMA-30B.

We apply weight selection to train small models on image classification datasets of different scales. We observe significant improvement in accuracy across datasets and models compared with baselines. Weight selection also substantially reduces the training time required to reach the same level of accuracy. Additionally, it can work alongside another popular method for knowledge transfer from large models – knowledge distillation (Hinton et al., 2015). We believe weight selection can be a general technique for training small models. Our work also encourages further research on utilizing pretrained models for efficient deployment.

## 2 RELATED WORK

**Weight initialization.** Weight initialization is a crucial aspect of model training. Glorot & Bengio (2010) maintain constant variance by setting the initial values of the weights using a normal distribution, aiming to prevent gradient vanishing or explosion. He et al. (2015) modify it to adapt to ReLU activations (Nair & Hinton, 2010). Mishkin & Matas (2016) craft the orthogonality in weight matrices to keep gradient from vanishing or exploding. Saxe et al. (2014) and Vorontsov et al. (2017) put soft constraints on weight matrices to ensure orthogonality.

There are methods that use external sources of knowledge like data distribution or unsupervised training to initialize neural networks. A data-dependent initialization can be obtained from performing K-means clustering or PCA (Krähenbühl et al., 2015; Tang et al., 2017) on the training dataset. Larochelle et al. (2009), Masci et al. (2011), Trinh et al. (2019), and Gani et al. (2022) show training on unsupervised objectives can provide a better initialization for supervised training.

**Utilizing pretrained models.** Transfer learning (Zhuang et al., 2020) is a common framework for using model weights pretrained from large-scale data. Model architecture is maintained and the model is fine-tuned on specific downstream tasks (Kolesnikov et al., 2020). Knowledge distillation involves training a usually smaller student model to approximate the output of a teacher model (Hinton et al., 2015; Tian et al., 2019; Beyer et al., 2022). This allows the student model to maintain the performance of a teacher while being computationally efficient. Another alternative approach for using pretrained models is through weight pruning (LeCun et al., 1990; Han et al., 2015; Li et al., 2017b; Liu et al., 2019). It involves removing less significant weights from the model, making it more efficient without significantly compromising performance.

Lin et al. (2020) transform parameters of a large network to an analogous smaller one through learnable linear layers using knowledge distillation to match block outputs. Sanh et al. (2019) and Shleifer & Rush (2020) create smaller models by initializing with a subset of layers from a pretrained BERT (Devlin et al., 2018). This method requires the smaller model to have the same width as teacher's. Czyzewski et al. (2022) borrow weights from existing convolutional networks to initialize novel convolutional neural networks. Chen et al. (2016) and Chen et al. (2021) transform weights from smaller networks as an effective initialization for larger models to accelerate convergence. Trockman et al. (2023) initialize convolutional layers with Gaussian distribution according to pretrained model's covariance. Similarly, Trockman & Kolter (2023) initialize self-attention layers according to observed diagonal patterns from pretrained ViTs. These two methods use statistics from but do not directly utilize pretrained parameters. The concurrent work, Xia et al. (2023), applies structured pruning to reduce existing large language models to their smaller versions and uses the pruned result as an initialization. Zhong et al. (2024) explores knowledge transfer between transformer-based LLM through parameters. Weight selection, in contrast, directly utilizes pretrained parameters, does not require extra computation, and is suitable for initializing any smaller variants of the pretrained model.

## 3  WEIGHT SELECTION

Given a pretrained model, our goal is to obtain an effective weight initialization for a smaller-size model within the same model family. Borrowing terminology from knowledge distillation, we refer to the pretrained model as *teacher* and the model we aim to initialize as *student*.

### 3.1  APPROACH

Modern neural network architectures often follow a modular approach: design a layer and replicate it to build the model (He et al., 2016; Vaswani et al., 2017; Tolstikhin et al., 2021; Dosovitskiy et al., 2021; Liu et al., 2022). This design ethos promotes scalability: models can be widened by increasing the embedding dimension or the number of channels in each block, and deepened by stacking more layers. It also enables us to perform weight selection following three steps: layer selection, component mapping, and element selection.

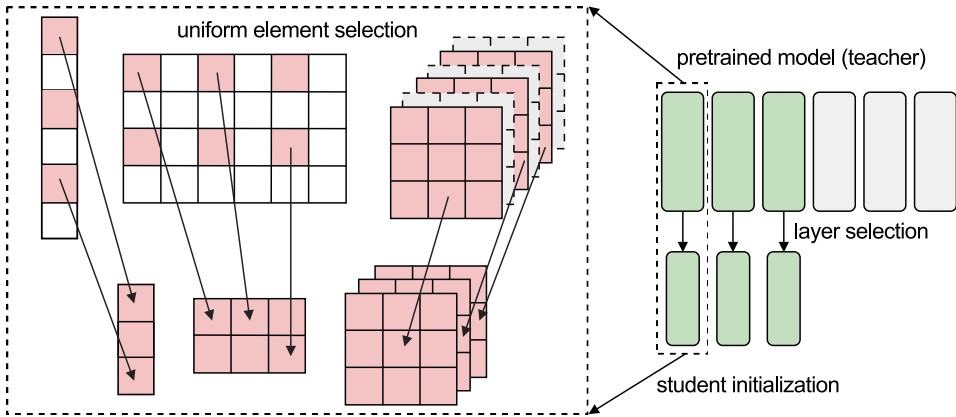

Figure 2: **Weight selection.** To initialize a smaller variant of a pretrained model, we uniformly select parameters from the corresponding component of the pretrained model.

**Layer selection.** Selecting layers from teacher is the first step. For each layer in student, a layer from teacher is selected as the source for initialization. The procedure for layer selection is slightly different for *isotropic* architectures and *hierarchical* architectures. An isotropic architecture refers to the neural network where each layer exhibits a consistent and uniform layerwise design throughout the model. ViT (Dosovitskiy et al., 2021) and MLP-mixer (Tolstikhin et al., 2021) belong to isotropic architectures. A hierarchical architecture is characterized by multi-scale representations and a hierarchy in embedding dimensions. Hierarchical architectures typically have stages with varying scales and embedding dimensions. For example, classic convolutional networks like VGG (Simonyan & Zisserman, 2014) progressively decrease spatial dimensions while increasing channel dimensions, capturing multi-scale features. Modern architectures like Swin-Transformer (Liu et al., 2021) and ConvNeXt (Liu et al., 2022) also employ this hierarchical design.

For isotropic architectures, we select the first $N$ layers from the teacher, where $N$ represents the student's layer count, denoted as *first-N selection*. When dealing with hierarchical structures like ConvNeXt (Liu et al., 2022), first-N selection is applied at each individual stage. An alternative method is *uniform* layer selection, where evenly-spaced layers in teacher are selected. Empirical results in Section 4.3 and Appendix B show that first-N layer selection is ideal for weight selection.

**Component mapping.** In the second step, we map components between student and teacher. From the previous step, we obtained layer mapping from teacher to student. The task is then reduced to initializing one student layer with one teacher layer. Thanks to the modular approach adopted by modern neural network design, layers in models of the same family have an identical set of components which only differ in their width. The process for matching the corresponding components is thus a natural one-to-one mapping.

**Element selection.** Upon establishing component mapping, the next step is to initialize student's component using its larger counterpart from teacher. The default method for element selection is

*uniform selection*, where evenly-spaced elements are selected from teacher's tensor as shown in figure 2. Details on *uniform selection* and other element selection methods will be introduced next part.

## 3.2 METHODS FOR ELEMENT SELECTION

In this part, we formulate element selection and introduce different selection criteria. Consider a weight tensor $W_s$ from student that we seek to initialize with teacher's weight tensor $W_t$. If $W_t$ has shape $t_1, t_2, ..., t_n$, then $W_s$, which is of the same component type with $W_t$, will also span n dimensions. Our goal is to select a subset of $W_t$'s elements to initialize $W_s$. Several possible methods for element selection are discussed as follows. We compare the performance of these element selection methods in Section 4.3. We find that as long as consistency is maintained (as explained in the paragraph *Random selection with consistency*, weight selection can achieve a similar level of performance. We propose using uniform selection as default for weight selection in practice.

**Uniform selection (default).** For each dimension $i$ of $W_t$, select evenly-spaced $s_i$ slices out of $t_i$. For example, to initialize a linear layer $W_s$ of shape $2 \times 3$ with a linear layer $W_t$ of shape $4 \times 6$, we select 1st and 3rd slice along the first dimension, and 1st, 3rd, and 5th slice along the second dimension. We present pseudocode for uniform selection in Algorithm 1. The algorithm starts with a copy of teacher's weight tensor $W_t$ and iteratively performs selection on all dimensions of $W_t$ to reach the desired shape for student. Notably, in architectures that incorporate grouped components — such as the multi-head attention module in ViTs and the grouped convolution in ResNeXt (Xie et al., 2017) — *uniform selection* absorbs information from all groups. For example, when applied to ViTs, *uniform selection* will select parameters from all heads in the attention block, which is likely to be beneficial for inheriting knowledge from the pretrained ViTs.

---

**Algorithm 1** Uniform element selection from teacher's weight tensor

---

**Input:** $W_t$          ▷ teacher's weight tensor
**Input:** $s$          ▷ desired dimension for student's weight tensor
**Output:** $W_s$ with shape $s$
 1: **procedure** UNIFORMELEMENTSELECTION($W_t, s$)
 2:      $W_s \leftarrow$ Copy of $W_t$          ▷ student's weight tensor
 3:      $n \leftarrow$ length of $W_t$.shape
 4:      **for** $i = 1 \rightarrow n$ **do**
 5:          $d_t \leftarrow W_t$.shape$[i]$
 6:          $d_s \leftarrow s[i]$
 7:          $indices \leftarrow$ Select $d_s$ evenly-spaced numbers from 1 to $d_t$
 8:          $W_s \leftarrow$ Select $indices$ along $W_s$'s $i^{th}$ dimension
 9:      **end for**
10:      **return** $W_s$
11: **end procedure**

---

**Consecutive selection.** For each dimension $i$ of $W_t$, select *consecutive* $s_i$ slices out of $t_i$. In contrast to *uniform selection*, for architectures with grouped components, *consecutive selection* selects some entire groups while omitting the contrast. For architectures without such grouped components, *consecutive selection* is equivalent to *uniform selection*.

**Random selection (with consistency).** For all weight tensors, and for each dimension $i$ of $W_t$, select the *same* randomly-generated set of $s_i$ slices out of $t_i$. Through empirical experiments in Section 4.3, we find that *consistency* (selecting the same indices for all weight matrices) is key for weight selection to reach its best performance. Motivation for maintaining consistency stems from the existence of residual connections — neurons that are added in the teacher model should have their operations preserved in the student. Furthermore, maintaining consistency preserves complete neurons during element selection, since only consistent positions are selected. It is worth noting that *uniform selection* and *consecutive selection* inherently preserve consistency, which are both special instances of *random selection with consistency*.

**Random selection (without consistency).** Along every dimension $i$ of $W_t$, randomly select $s_i$ slices out of $t_i$. Unlike *random selection w/ consistency*, this method does not require selecting the same indices for every weight tensor. We design this method to examine the importance of *consistency*.

## 4 EXPERIMENTS

### 4.1 SETTINGS

**Datasets.** We evaluate weight selection on 9 image classification datasets including ImageNet-1K (Deng et al., 2009), CIFAR-10, CIFAR-100 (Krizhevsky, 2009), Flowers (Nilsback & Zisserman, 2008), Pets (Parkhi et al., 2012), STL-10 (Coates et al., 2011), Food-101 (Bossard et al., 2014)), DTD (Cimpoi et al., 2014), SVHN (Netzer et al., 2011) and EuroSAT (Helber et al., 2019; 2018). These datasets vary in scales ranging from 5K to 1.3M training images.

**Models.** We perform experiments on ViT-T/16 (Touvron et al., 2021a) and ConvNeXt-F (Liu et al., 2022), with ImageNet-21K pretrained ViT-S/16 and ConvNeXt-T as their teachers respectively. We obtain weights for ImageNet-21K pretrained ViT-S/16 from Steiner et al. (2021) and ImageNet-21K pretrained ConvNeXt-T from Liu et al. (2022). We present the detailed configurations in Table 1.

| configuration | student | | teacher | |
|---|---|---|---|---|
| model | ViT-T | ConvNeXt-F | ViT-S | ConvNeXt-T |
| depth | 12 | 2 / 2 / 6 / 2 | 12 | 3 / 3 / 9 / 3 |
| embedding dimension | 192 | 96 / 192 / 384 / 768 | 384 | 48 / 96 / 192 / 384 |
| number of heads | 3 | - | 6 | - |
| number of parameters | 5M | 5M | 22M | 28M |

Table 1: **Model Configurations.** We perform main experiments on ConvNeXt and ViT, and use student that halve the embedding dimensions of their corresponding teacher.

**Training.** We follow the training recipe from ConvNeXt (Liu et al., 2022) with adjustments to batch size, learning rate, and stochastic depth rate (Huang et al., 2016) for different datasets. See Appendix A for details. To ensure a fair comparison, we adapt hyperparameters to baseline (training with random initialization), and the same set of hyperparameters is used for training with weight selection.

**Random initialization baseline.** We utilize the model-specific default initialization from timm library (Wightman, 2019), a popular computer vision library with reliable reproducibility. Its default initialization of ViT-T and ConvNeXt-F employs a truncated normal distribution with a standard deviation of 0.02 for linear and convolution layers. The truncated normal distribution, designed to clip initialization values, is adopted to develop modern neural networks (Liu et al., 2022).

### 4.2 RESULTS

Experiment results are presented in Table 2. Across all 9 image classification datasets, weight selection consistently boosts test accuracy, especially for smaller datasets. Notably, weight selection addresses the well-known challenge of training ViT on small datasets, which likely contributes to the significant accuracy improvement for ViT. Training curves for ImageNet-1K are shown in Figure 3. Both models benefit from weight selection early on and maintain this advantage throughout training.

| dataset (scale ↓) | random init | weight selection | change | random init | weight selection | change |
|---|---|---|---|---|---|---|
| ImageNet-1K | 73.9 | 75.6 | ↑1.6 | 76.1 | 76.4 | ↑0.3 |
| SVHN | 94.9 | 96.5 | ↑1.6 | 95.7 | 96.9 | ↑1.2 |
| Food-101 | 79.6 | 86.9 | ↑7.3 | 86.9 | 89.0 | ↑2.1 |
| EuroSAT | 97.5 | 98.6 | ↑1.1 | 98.4 | 98.8 | ↑0.4 |
| CIFAR-10 | 92.4 | 97.0 | ↑4.6 | 96.6 | 97.4 | ↑0.8 |
| CIFAR-100 | 72.3 | 81.4 | ↑9.1 | 81.4 | 84.4 | ↑3.0 |
| STL-10 | 61.5 | 83.4 | ↑21.9 | 81.4 | 92.3 | ↑10.9 |
| Flowers | 62.4 | 81.9 | ↑19.5 | 80.3 | 94.5 | ↑14.2 |
| Pets | 25.0 | 68.6 | ↑43.6 | 72.9 | 87.3 | ↑14.4 |
| DTD | 49.4 | 62.5 | ↑13.1 | 63.7 | 68.8 | ↑5.1 |
| | (a) ViT-T | | | (b) ConvNeXt-F | | |

Table 2: **Test accuracy on image classification datasets.** On all 9 datasets, employing weight selection for initialization leads to an improvement in test accuracy. Datasets are ordered by their image counts. Weight selection provides more benefits when evaluated on datasets with fewer images.

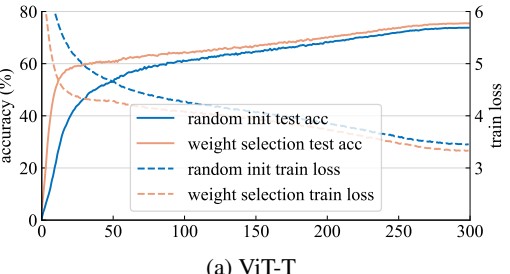 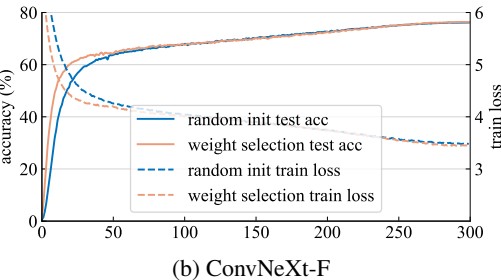

(a) ViT-T  (b) ConvNeXt-F

Figure 3: **Training curves on ImageNet-1K.** When initialized using weight selection from ImageNet-21K pretrained models, both ViT-T (from ViT-S) and ConvNeXt-F (from ConvNeXt-T) exhibit superior performance compared to their randomly-initialized counterparts.

### 4.3 COMPARISONS

**Advantage over classic initialization.** We compare weight selection and its variants with two widely-adopted initialization methods: Xavier initialization (Glorot & Bengio, 2010) and Kaiming initialization (He et al., 2015) and present results on CIFAR-100 as shown in Table 3. All variants of weight selection yield considerably better results than classic initialization methods.

**Comparison of selection methods.** Consistency is the key for weight selection to reach its best performance. For both model architectures, uniform selection, consecutive selection, and random selection with consistency achieve a similar level of performance. Note that uniform selection and consecutive selection inherently maintain consistency. When removing consistency, a sharp drop in performance is observed. For random selection experiments, we report the median performance of three different selection results.

| init | ViT-T | ConvNeXt-F |
|---|---|---|
| timm default (trunc normal) | 72.3 | 81.4 |
| Xavier (Glorot & Bengio, 2010) | 72.1 | 82.8 |
| Kaiming (He et al., 2015) | 73.0 | 82.5 |
| weight selection (uniform) | 81.4 | **84.4** |
| weight selection (consecutive) | 81.6 | 84.0 |
| weight selection (random w/ consistency) | **81.7** | 83.9 |
| weight selection (random w/o consistency) | 77.4 | 82.8 |

Table 3: **Comparison with classic initialization methods.** We present CIFAR-100 test accuracy for weight selection's variants and classic initialization methods. Weight selection methods with consistency outperform classic initialization methods by a large margin.

### 4.4 COMPATIBILITY WITH KNOWLEDGE DISTILLATION

Weight selection transfers knowledge from pretrained models via parameters. Another popular approach for knowledge transferring is knowledge distillation (Hinton et al., 2015), which utilizes outputs from pretrained models. Here we explore the compatibility of these two techniques.

**Settings.** We evaluate the performance of combining weight selection with two different approaches in knowledge distillation – logit-based distillation and feature-based distillation by using weight selection as initialization for student in knowledge distillation. Logit-based distillation uses KL-divergence as the loss function for matching student's and teacher's logits. Denote student's output probabilities as $p_s$, and teacher's output probabilities as $p_t$, the loss for logit-based distillation can be formulated as

$$\mathcal{L} = \mathcal{L}_{class} + \alpha \cdot KL(p_t || p_s) \tag{1}$$

where $L_{class}$ is supervised loss, and $\alpha$ is the coefficient for distillation loss. Note that matching logits requires teacher to be trained on the same dataset as student. For logit-based distillation, We train ViT-T on ImageNet-1K while using the ImageNet-1K pretrained ViT-S model from DeiT (Touvron et al., 2021a) as the teacher for both knowledge distillation and weight selection. $\alpha$ is set to 1.

Feature-based distillation steps in when a classification head of the target dataset is not available. Denote teacher's output as $O_t$, and student's as $O_s$. Feature-based distillation can be formulated as

$$\mathcal{L} = \mathcal{L}_{class} + \alpha \cdot L_1(O_t, MLP(O_s)) \tag{2}$$

An MLP is used to project student's output to teacher's embedding dimension, and $L_1$ loss is used to match the projected student's output and teacher's output. For feature-based distillation, we perform CIFAR-100 training experiments on ViT-T, using ImageNet-21K pretrained ViT-S as the teacher for both knowledge distillation and weight selection. We tune $\alpha$ on distillation trials and use the same value for $\alpha$ for trials that combine distillation and weight selection.

**Results.** Table 4 provides results for knowledge distillation and weight selection when applied individually or together. These results show the compatibility between weight selection and different types of knowledge distillation. Without incurring additional inference costs, employing weight selection alone produces a better result than vanilla logit-based distillation and feature-based distillation. More importantly, the combination of weight selection and knowledge distillation delivers the best results, boosting accuracies to 76.0% on ImageNet-1K and 83.9% on CIFAR-100. These results further confirm weight selection's usefulness as an independent technique and the compatibility between weight selection and knowledge distillation.

| setting | ImageNet-1K (logit-based distillation) | | CIFAR-100 (feature-based distillation) | |
|---|---|---|---|---|
| | test acc | change | test acc | change |
| baseline | 73.9 | - | 72.3 | - |
| distill | 74.8 | ↑0.9 | 78.4 | ↑6.4 |
| weight selection | **75.5** | ↑1.6 | **81.4** | ↑9.1 |
| distill + weight selection | **76.0** | ↑2.1 | **83.9** | ↑11.6 |

Table 4: **Compatibility with knowledge distillation.** Weight selection is useful as an independent technique, and can be combined with knowledge distillation to achieve the best performance.

## 5 ANALYSIS

In this section, we perform a comprehensive analysis of weight selection. Unless otherwise specified, the standard setting (for comparison) of weight selection is initializing ViT-T with uniform selection from the ImageNet-21K pretrained ViT-S, trained and evaluated on CIFAR-100.

**Reduction in training time.** We find weight selection can significantly reduce training time. We directly measure the reduction in training time by training ViT-T with weight selection for different numbers of epochs and present the results in Figure 4a. The number of warmup epochs is modified to maintain its ratio with the total epochs for each trial. With weight selection, the same performance on CIFAR-100 can be obtained with only 1/3 epochs compared to training from random initialization.

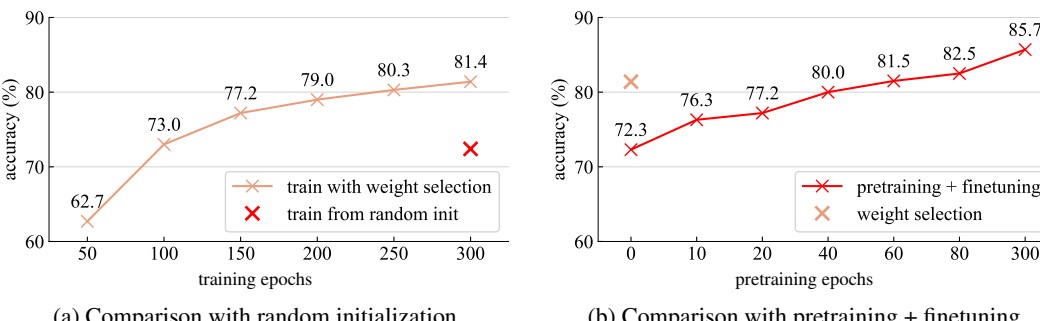

(a) Comparison with random initialization      (b) Comparison with pretraining + finetuning

Figure 4: **Faster training.** Compared to random initialization, ViT-T can reach the same performance on CIFAR-100 with only 1/3 epochs compared to training from random initialization. When compared to pretraining (on ImageNet-1K) + finetuning, weight selection is able to match the accuracy at 60 epochs of pretraining, saving 6.12x training time.

**Comparison with transfer learning.** We conduct experiments to find the training budget needed for pretraining to match the accuracy of weight selection. For this experiment, we train ViT-T on

ImageNet-1K with different numbers of epochs and then fine-tune on CIFAR-100 for 300 epochs. As shown in Figure 4b, it takes 60 epochs of pretraining on ImageNet-1K to achieve the same performance on CIFAR-100. Under this setting, weight selection is 6.12x faster compared to reaching the same performance with pretraining, without need to access the dataset used for pretraining.

**Pretrained models.** We study the effect of using different pretrained models as weight selection teacher. Models with supervised pretraining turn out to be the best teacher. We evaluate the performance of ViT-B as teacher under different pretraining regimes: CLIP (Radford et al., 2021), MAE (He et al., 2022), and DINO (Caron et al., 2021). Table 5 presents the results. Initializing with pretrained weights consistently outperforms random initialization. ImageNet-21K pretrained teacher provides the most effective initialization. Note that for this experiment, we use ViT-B as teacher for weight selection, since it is the smallest model that MAE and CLIP provide.

| Pretrained models | CIFAR-10 | CIFAR-100 | STL-10 |
|---|---|---|---|
| supervised (ImageNet-21K) | 95.1 | **77.6** | **73.1** |
| CLIP (Radford et al., 2021) | 94.9 | 77.3 | 66.0 |
| MAE (He et al., 2022) | **95.9** | 77.2 | 71.0 |
| DINO (Caron et al., 2021) | 95.0 | 75.7 | 69.4 |

Table 5: **Different pretrained models.** Supervised pretraining makes the best teacher.

**Layer selection.** Shleifer & Rush (2020) select evenly-spaced layers from BERT to initialize small models. We compare two layer selection methods: first-N layer selection and uniform layer selection. To evaluate different layer selection methods, we create ViT-A of 6 layers (half of the ViT-T depth), with other configurations identical to ViT-T. In this experiment, we use ViT-A and ConvNeXt-F as student, and ImageNet-21K pretrained ViT-S and ConvNeXt-T as their weight selection teacher. From the results presented in Table 6, we find first-N layer selection performs consistently better than uniform layer selection. Presumably, since layers initialized by first-N selection are naturally contiguous and closer to input processing, they offer a more effective initialization for smaller models.

| setting | ViT-A | ConvNeXt-F |
|---|---|---|
| random init | 69.6 | 81.3 |
| first-N layer selection | **77.6** | **84.4** |
| uniform layer selection | 76.7 | 83.2 |

Table 6: **Layer selection.** First-N layer selection performs better than uniform layer selection.

**Comparison with pruning.** We test the existing structured and unstructured pruning methods on reducing pretrained ViT-S to ViT-T. An important thing to note is that our setting is different from neural network pruning. Structured pruning (Li et al., 2017a) typically only prunes within residual blocks for networks with residual connections, and unstructured pruning (Han et al., 2015) prune weights by setting weights to zero instead of removing it. Despite that these pruning methods are not designed for our setting, we can extend structured and unstructured pruning methods to be applied here. Specifically, we can adopt $L_1$ pruning and magnitude pruning for element selection. For $L_1$ pruning, we use $L_1$ norm to prune the embedding dimension. For magnitude pruning, we squeeze the selected parameters into the required shape of student.

We present results in Table 7. $L_1$ *pruning* yields better results compared to random initialization baseline. However, since $L_1$ norm inevitably breaks consistency, it could not reach the same performance with weight selection. *Magnitude pruning* only produces marginally better results over random initialization, presumably due to the squeezing operation which breaks the original structure.

| setting | ViT-T | ConvNeXt-F |
|---|---|---|
| random init | 72.3 | 81.4 |
| weight selection | **81.4** | **84.4** |
| $L_1$ pruning | 79.5 | 82.8 |
| magnitude pruning | 73.8 | 81.9 |

Table 7: **Comparison with pruning.** $L_1$ and *magnitude* pruning performs worse than weight selection.

| teacher | params | test acc |
|---|---|---|
| ViT-S | 22M | **81.4** |
| ViT-B | 86M | 77.6 |
| ViT-L | 307M | 76.9 |

Table 8: **Teacher's size.** Smaller teacher provides better initialization.

**Teacher's size.** We show that initializing from a teacher of closer size produces better results. A larger teacher means a higher percentage of parameters will be discarded, which translates to more

information loss during weight selection. We present results for using ViT-S, ViT-B, and ViT-L as weight selection teacher to initialize ViT-T in Table 8. Interestingly, even a 5M-parameter subset from 301M parameters in ViT-L is an effective initialization, increasing accuracy by 4.5%.

**Linear probing.** We use linear probing to directly measure the raw model's ability as a feature extractor, which can be a good indicator of the initialization quality. Linear probing is a technique used to assess a pretrained model's representations by training a linear classifier on top of the fixed features extracted from the model.

Following the recipe in He et al. (2022), we apply linear probing on CIFAR-100 to evaluate ViT-T and ConvNeXt-F initialized with weight selection from their ImageNet-21K pretrained teachers, ViT-S and ConvNeXt-T respectively. We compare between random initialization and weight selection as shown in Table 9. Without any training, weight selection performs significantly better than random initialization in producing features.

| setting | ViT-T | ConvNeXt-F |
|---|---|---|
| random init | 13.5 | 7.1 |
| weight selection | **28.2** | **23.6** |

Table 9: **Linear probing on CIFAR-100.** Weight selection produces a non-trivial feature extractor.

**Mimetic initialization.** Mimetic initialization (Trockman & Kolter, 2023) uses the diagonal properties observed in the pretrained self-attention layer's weights to initialize ViTs. We present results for mimetic initialization in Table 10. By directly utilizing pretrained parameters, weight selection outperforms mimetic initialization by a large margin. In addition, we visualize the product of $W_q W_k^T$ and $V W_{proj}$ the first head in the first attention block of ViT-T with random initialization, pretrained ViT-S, and ViT-T with weight selection. As shown in Figure 5, weight selection enables small models to inherit the desirable diagonal properties in their self-attention layers, which typically only exist in pretrained models.

| setting | CIFAR-10 | CIFAR-100 | STL-10 |
|---|---|---|---|
| random init | 92.4 | 72.3 | 61.5 |
| mimetic init | 93.3 | 74.7 | 67.5 |
| weight selection | **97.0** | **81.4** | **83.4** |

Table 10: **Comparison with mimetic initialization.** Weight selection significantly outperforms mimetic initialization by directly utilizing pretrained parameters.

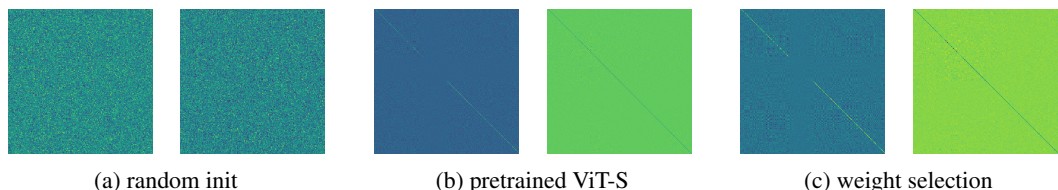

    (a) random init            (b) pretrained ViT-S           (c) weight selection

Figure 5: **Visualization of self-attention layers.** Visualization of $W_q W_k^T$ (left) and $V W_{proj}$ (right) for ViT-T with random initialization, pretrained ViT-S, and ViT-T with weight selection. Weight selection can inherit the diagonal property of self-attention layers that only exists in pretrained ViTs.

## 6 CONCLUSION

We propose weight selection, a novel initialization method that utilizes large pretrained models. With no extra cost, it is effective for improving the accuracy of a smaller model and reducing its training time needed to reach a certain accuracy level. We extensively analyze its properties and compare it with alternative methods. We hope our research can inspire further exploration into utilizing large pretrained models to create smaller models.

**Acknowledgement.** We would like to thank Haodi Zou for creating illustrative figures, and Mingjie Sun, Boya Zeng, Chen Wang, Jiatao Gu, Maolin Mao for valuable discussions and feedback.

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

# APPENDIX

## A   TRAINING SETTINGS

**Training recipe.** We provide our training recipe with configurations in Table 11. The recipe is adapted from ConvNeXt (Liu et al., 2022).

| Training Setting | Configuration |
|---|---|
| optimizer | AdamW |
| base learning rate | 4e-3 |
| weight decay | 0.05 |
| optimizer momentum | $\beta_1, \beta_2 = 0.9, 0.999$ |
| batch size | 4096 |
| training epochs | 300 |
| learning rate schedule | cosine decay |
| warmup epochs | 50 |
| warmup schedule | linear |
| layer-wise lr decay (Clark et al., 2020; Bao et al., 2022) | None |
| randaugment (Cubuk et al., 2020) | (9, 0.5) |
| mixup (Zhang et al., 2018) | 0.8 |
| cutmix (Yun et al., 2019) | 1.0 |
| random erasing (Zhong et al., 2020) | 0.25 |
| label smoothing (Szegedy et al., 2016) | 0.1 |
| layer scale (Touvron et al., 2021b) | 1e-6 |
| head init scale (Touvron et al., 2021b) | None |
| gradient clip | None |

Table 11: **Our basic recipe.**

**Hyper-parameters.** Table 12 and Table 13 record batch size, warmup epochs, and training epochs of ConvNeXt-F and Vit-T, respectively, for each dataset. The batch size of each dataset is chosen proportional to its total size. The warmup epochs are set as around one-fifth of the total training epochs. Base learning rates for ConvNeXt-F and ViT-T are 4e-3 and 2e-3 respectively.

| | C-10 | C-100 | Pets | Flowers | STL-10 | Food101 | DTD | SVHN | EuroSAT | IN1k |
|---|---|---|---|---|---|---|---|---|---|---|
| batch size | 1024 | 1024 | 128 | 128 | 128 | 1024 | 128 | 1024 | 512 | 4096 |
| warmup epochs | 50 | 50 | 100 | 100 | 50 | 50 | 100 | 10 | 50 | 50 |
| training epochs | 300 | 300 | 600 | 600 | 300 | 300 | 600 | 50 | 300 | 300 |
| drop path rate | 0.1 | 0.1 | 0.1 | 0.1 | 0 | 0.1 | 0.2 | 0.1 | 0.1 | 0 |

Table 12: **Hyper-parameter setting on ConvNeXt-F.**

| | C-10 | C-100 | Pets | Flowers | STL-10 | Food101 | DTD | SVHN | EuroSAT | IN1k |
|---|---|---|---|---|---|---|---|---|---|---|
| batch size | 512 | 512 | 512 | 512 | 512 | 512 | 512 | 512 | 512 | 4096 |
| warmup epochs | 50 | 50 | 100 | 100 | 50 | 50 | 100 | 10 | 50 | 50 |
| training epochs | 300 | 300 | 600 | 600 | 300 | 300 | 600 | 50 | 300 | 300 |

Table 13: **Hyper-parameter setting on ViT-T.**

## B    ADDITIONAL ANALYSIS

**Layer selection.** To further confirm the first-N layer selection's superiority, we conduct experiments that rule out the effect of element selection. Specifically, we evaluate the results of directly fine-tuning different subsets of 6 layers out of 12 layers of a pretrained ViT-T model. We present the results in Table 15. There is a clear trend favoring shallow layers, with first-N layers better than mid-N layers, and both of them have better performance than last-N layers. Uniform layer selection is slightly worse than first-N layer selection. Based on this empirical result, we find first-N layer selection a suitable method for weight selection.

| setting | CIFAR-100 test acc |
|---|---|
| first-N layer selection | **81.6** |
| mid-N layer selection | 68.3 |
| last-N layer selection | 62.0 |
| uniform layer selection | 76.3 |

Table 14: **Layer selection.** First-N layer selection performs significantly better than uniform layer selection when ruling out the effect of element selection.

We also find that empirical experiments from ViT-L (which has 24 layers) to ViT-S has different preference over layer selection. Under this setting, uniform layer selection achieves slightly better performance than first-N layer selection. Our main approach would still adopt first-N layer selection as the default method, for the reason that in the case of a smaller teacher (which is favored in weight selection), first-N layer selection consistently outperforms its alternatives.

| setting | CIFAR-100 test acc |
|---|---|
| first-N layer selection | 76.9 |
| mid-N layer selection | 75.9 |
| last-N layer selection | 77.1 |
| uniform layer selection | **77.5** |

Table 15: **Layer selection (ViT-L as teacher).** Uniform layer selection yields slightly better results than first-N layer selection when student's ratio to teacher is small.

**Weight components.** We conduct ablation studies on ViT-T to understand the influence of distinct model components on performance. In particular, we evaluate the performance of weight selection without one of the following particular types of layers: patch embedding, position embedding, attention block, normalization layer, or MLP layer. As illustrated in Table 16, excluding component from initialization leads to substantial drops in accuracy for all datasets. The results confirm that initializing with all components from pretrained models is necessary.

| Setting | CIFAR-10 | CIFAR-100 | STL-10 |
|---|---|---|---|
| random init | 92.4 | 72.3 | 61.5 |
| weight selection | **97.0** | **81.4** | **83.4** |
| w/o patch embed | 96.8 | 79.5 | 77.1 |
| w/o pos embed | 95.6 | 78.4 | 80.2 |
| w/o attention | 96.2 | 77.3 | 80.5 |
| w/o normalization | 96.2 | 79.0 | 79.8 |
| w/o mlp | 95.6 | 78.8 | 74.2 |

Table 16: **ViT component ablation.** Using all components from pretrained models is the best.

**Longer training on ImageNet-1K.** To assess if weight selection remains beneficial for extended training durations, we use the improved training recipe from Liu et al. (2023). Specifically, the total epochs are extended to 600, and mixup / cutmix are reduced to 0.3. The results, as displayed in Table 17, affirm that our method continues to provide an advantage even under extended training durations. Both ViT-T and ConvNeXt-F, when initialized using weight selection, consistently surpass models with random initialization. This confirms that weight selection does not compromise the model's capacity to benefit from longer training.

| setting | ViT-T | | ConvNeXt-F | |
|---|---|---|---|---|
| | test acc | change | test acc | change |
| random init | 73.9 | - | 76.1 | - |
| weight selection | **75.5** | ↑1.6 | **76.4** | ↑0.3 |
| random init (longer training) | 76.3 | - | 77.5 | - |
| weight selection (longer training) | **77.4** | ↑1.1 | **77.7** | ↑0.2 |

Table 17: **Longer training.** Weight selection's improvement is robust under stronger recipe.

## C  RESULTS ON MORE ARCHITECTURES

We present results on additional architectures, namely ResNet (He et al., 2016) and MLP-mixer (Tolstikhin et al., 2021). Model configurations are provided in Table 18 and the detailed results on image classification datasets are presented in 19. Notably, MLP-mixer enjoys more benefits from weight selection. We speculate that it is due to different inductive bias among different architectures.

| configuration | student | | teacher | |
|---|---|---|---|---|
| model | Mixer-b16-half | ResNet-18 | Mixer-b16 | ResNet-34 |
| depth | 12 | 2 / 2 / 2 / 2 | 12 | 3 / 4 / 6 / 3 |
| embedding dimension | 384 | 64 / 128 / 256 / 512 | 768 | 64 / 128 / 256 / 512 |
| number of parameters | 15.8M | 11M | 59.9M | 63.5M |

Table 18: **Model Configurations.** We perform experiments on Mixer-MLP and ResNet, and use student that halve the embedding dimensions of their corresponding teacher.

| dataset (scale ↓) | random init | weight selection | change | random init | weight selection | change |
|---|---|---|---|---|---|---|
| SVHN | 95.9 | 96.2 | ↑0.3 | 87.9 | 93.3 | ↑5.4 |
| Food-101 | 83.8 | 85.9 | ↑2.1 | 77.5 | 83.7 | ↑6.2 |
| EuroSAT | 97.5 | 98.6 | ↑1.1 | 98.4 | 98.8 | ↑0.4 |
| CIFAR-10 | 96.4 | 97.1 | ↑0.7 | 93.5 | 96.1 | ↑2.6 |
| CIFAR-100 | 80.3 | 82.5 | ↑2.2 | 73.4 | 79.8 | ↑6.4 |
| STL-10 | 84.3 | 94.1 | ↑9.8 | 73.3 | 82.0 | ↑8.7 |
| Flowers | 78.1 | 95.8 | ↑17.7 | 69.1 | 83.2 | ↑14.1 |
| Pets | 78.2 | 87.8 | ↑9.6 | 50.4 | 69.6 | ↑19.2 |
| DTD | 49.5 | 64.6 | ↑15.1 | 42.9 | 49.3 | ↑6.4 |
| | (a) ResNet | | | (b) MLP-mixer | | |

Table 19: **Test accuracy on image classification datasets.** Weight selection shows consistent improvements on other architectures across all 9 image classification datasets.

