# OpenReview forum: "Initializing Models with Larger Ones"
_ICLR.cc/2024/Conference — ICLR 2024 spotlight_

### Official Review · Reviewer_FWr5 · 2023-10-30

**Soundness:** 2 fair
**Presentation:** 2 fair
**Contribution:** 3 good
**Rating:** 6
**Confidence:** 4

**Summary:**

The paper suggests that initializing the weight parameters of a smaller neural network from a pre-trained, larger model can significantly improve the smaller model's performance.

**Strengths:**

- This paper is easy to follow;
- The proposed method can improve the performance.

**Weaknesses:**

- The proposed method of selecting layers or weights for initialization is somewhat trivial. A more technical approach is encouraged to fully leverage the capabilities of larger models.

- The study is limited in its scope by only employing ViT-T and ConvNeXt-F architectures. Inclusion of additional architectures such as Swin, PVT, and EfficientViT would enrich the evaluation.

- There is an inconsistency in the reported accuracies for the same ImageNet-1k on the same model across different tables. For instance, ConvNeXt-F yields an accuracy of 76.1% in Table 1 but achieves around 82.0% in Table 2. Clarification is needed.

- A comparative analysis involving ConvNeXt is crucial in the "COMPATIBILITY WITH KNOWLEDGE DISTILLATION" section, given its architectural differences with ViT.

- The paper lacks sufficient motivation and theoretical grounding for the proposed selection strategy. More insights are needed to justify why, for example, the first N layers are selected, and how this methodology generalizes to other architectures.

- For better comprehension, it would be beneficial to present the customized structures, such as ConvNeXt-F, in tabulated form.

- The potential for leveraging the weights of larger models (which are more powerful) is high. However, Table 7 indicates that the proposed method struggles to effectively learn from these larger models.

- Some grammar errors:
	- "Glorot & Bengio (2010a) maintains " -> "Glorot & Bengio (2010a) maintain";
	- "Lin et al. (2020) transforms" -> "Lin et al. (2020) transform";
	- etc.

**Questions:**

Please see Section "Weaknesses".

---

> ### Author Response · Authors · 2023-11-22
> **Rebuttal by Authors**
>
> We sincerely thank you for your constructive comments. We are encouraged that you find our paper easy to follow. We would like to address the comments and questions below:
>
> > W1: The proposed method of selecting layers or weights for initialization is somewhat trivial. A more technical approach is encouraged to fully leverage the capabilities of larger models.
> > W5: The paper lacks sufficient motivation and theoretical grounding for the proposed selection strategy.
>
> Weight selection is designed to be easily adopted by people who want to use pretrained models directly without incurring any additional computation costs. We find that uniform selection is a cost-free approach that contains desirable properties (maintaining consistency) in the parameter selection process, and performs better than pruning approaches (as shown in Section 5: comparisons with pruning).
>
> **Justification for uniform element selection**
>
> In the context of transforming larger models into smaller models involving a change in embedding dimension, we identify **maintaining consistency (selecting the same indices for all weight matrices)** as the critical factor. Specifically, for all networks with residual connections, the layer can be represented mathematically as:
>
> $$ y = F(x) + x $$
>
> where $x, y \in \mathbb{R}^D$, and $D$ is the embedding dimension. This equation can also be expressed element-wise:
>
> $$ y_i = F(x)_i + x_i$$
>
> for all i from 0 to $D - 1$.
>
> Traditional frameworks that reduce model sizes (pruning) do not alter D, and consequently, they do not directly modify this equation. Under our setting, which reduces D to a smaller value, selecting the same set of indices on the input neurons and along F(⋅) (maintaining consistency) will preserve the addition operations on the subset of the original residual connection operations.
>
> As demonstrated in Table 3, random selection with consistency outperforms random selection without consistency by a significant margin (an accuracy improvement of 4%). Moreover, random selection with consistency exhibits comparable performance to uniform/consecutive selection. Therefore, we determine that **maintaining consistency is the factor that contributes most of the performance.** To simplify and standardize weight selection in practice, we adopt uniform element selection (which inherently preserves consistency) as the default method for weight selection.
>
> **Justification for layer selection**
>
> Our ablation on layer selection confirms the superiority of first-N layer selection as shown in Table 6. We present Table 6 below for reference. As illustrated in section 5, to examine the effect of layer selection, we create Vit-A, a variant of ViT with 6 layers, with other model configurations identical to ViT-T. We present weight selection results (from ViT-S to ViT-A, from ConvNeXt-T to ConvNeXt-F) with different layer selection methods.
>
> | Setting / Model                            | ViT-A | ConvNeXt-F |
> | ------------------------------------------ |:-----:|:----------:|
> | random init                                | 69.6  |    81.3    |
> | first-N layer selection                    | 77.6  |    84.4    |
> | maximally-spaced / uniform layer selection | 76.7  |    83.2    |
>
> To rule out the effect of element selection, we present results of directly finetuning different subsets of 6 layers out of 12 layers of a pretrained ViT-T model. We present the results as follows:
>
> | layer selection            | CIFAR-100 test acc |
> | -------------------------- |:------------------:|
> | first-N                    |        81.6        |
> | mid-N                      |        68.3        |
> | last-N                     |        62.0        |
> | maximally-spaced / uniform |        76.3        |
>
> We observe a clear trend favoring shallow layers, with first-N > middle-N > last-N. And uniform layer selection is slightly worse than first-N layer selection. Based on these empirical results from two experiments listed above, we find first-N layer selection a suitable method in weight selection. We have included this analysis for layer selection in Appendix B.

---

> ### Author Response · Authors · 2023-11-22
> **Rebuttal by Authors**
>
> > W2: The study is limited in its scope by only employing ViT-T and ConvNeXt-F architectures. Inclusion of additional architectures such as Swin, PVT, and EfficientViT would enrich the evaluation.
>
> We present results on weight selection on **ResNet [He2016], MLP-Mixer [Tolstikhin2021], Swin Transformer [Liu2021], and Pyramid Vision Transformer [Wang2021]**, evaluated on both CIFAR-10 and CIFAR-100, along with the respective model configuration for the student and the teacher.
>
> **Model Configuration**
>
> |        Architecture        |  student   | # of parameters |  teacher   | # of parameters |
> |:--------------------------:|:----------:|:---------------:|:----------:|:---------------:|
> |           ResNet           | ResNet-18  |      11.7M      | ResNet-34  |      21.8M      |
> |         Mlp-Mixer          | Mixer-T/32 |      5.4M       | Mixer-S/32 |      19.1M      |
> |      Swin-Transformer      |   Swin-F   |      7.53M      |   Swin-T   |      28.5M      |
> | Pyramid Vision transformer | PVT-v2-b0  |      3.7M       | PVT-v2-b1  |      14.0M      |
>
>
> **CIFAR-10**
> | Setting / Model  | ResNet    | Mlp-Mixer | Swin Transformer | PVT  |
> | ---------------- | --- |:---------:|:----------------:|:----:|
> | random init      |   96.4  |   90.8    |       94.9       | 96.3 |
> | weight selection |   97.1  |   95.1    |       96.5       | 97.4 |
>
> **CIFAR-100**
> | Setting / Model  | ResNet    | Mlp-Mixer | Swin Transformer | PVT  |
> | ---------------- | --- |:---------:|:----------------:|:----:|
> | random init      |   80.3  |   72.3    |       79.0       | 81.5 |
> | weight selection |   82.3  |   77.9    |       81.7       | 83.4 |
>
> Over various architectures and datasets, we find weight selection provides consistently better results than random initialization. Interestingly, for architectures with less inductive bias like MLP-mixer and ViT(included in the main results), a higher increase in performance is observed.
>
> [He2016] He, Kaiming, et al. "Deep residual learning for image recognition." Proceedings of the IEEE conference on computer vision and pattern recognition. 2016.
>
> [Tolstikhin2021] Tolstikhin, Ilya O., et al. "Mlp-mixer: An all-mlp architecture for vision." Advances in neural information processing systems 34 (2021): 24261-24272.
>
> [Liu2021] Liu, Ze, et al. "Swin transformer: Hierarchical vision transformer using shifted windows." Proceedings of the IEEE/CVF international conference on computer vision. 2021.
>
> [Wang2021] Wang, Wenhai, et al. "Pyramid vision transformer: A versatile backbone for dense prediction without convolutions." Proceedings of the IEEE/CVF international conference on computer vision. 2021.
> > W3: There is an inconsistency in the reported accuracies for the same ImageNet-1k on the same model across different tables. For instance, ConvNeXt-F yields an accuracy of 76.1% in Table 1 but achieves around 82.0% in Table 2. Clarification is needed.
>
> We present CIFAR-100 test accuracy for this experiment. We have edited paper to clarify that.
>
> > W4: A comparative analysis involving ConvNeXt is crucial in the "COMPATIBILITY WITH KNOWLEDGE DISTILLATION" section, given its architectural differences with ViT.
>
> | ConvNeXt-F                 | CIFAR-100 test accuracy |
> | -------------------------- | :---------------------: |
> | baseline (random init)     |          81.4           |
> | weight selection           |          84.4           |
> | distill                    |          82.1           |
> | weight selection + distill |          85.0           |
>
> We can see a similar trend on ConvNeXt as the results of ViT shown in Table 4. These results further confirm weight selection's usefulness as an independent technique and the compatibility between weight selection and knowledge distillation, on both ViTs and ConvNets.

---

> ### Author Response · Authors · 2023-11-22
> **Rebuttal by Authors**
>
> > W6: For better comprehension, it would be beneficial to present the customized structures, such as ConvNeXt-F, in tabulated form.
>
> Thanks, we have included model configurations for ViT-T, ViT-S, ConvNeXt-F, and ConvNeXt-T in Table 1. For reference, we also provide the model configuration of ConvNeXt-F and ConvNeXt-T here.
>
> ConvNeXt-T: depth: (3,3,9,3) embedding dimension: (96, 192, 384, 768)
> ConvNeXt-F: depth: (2,2,6,2) embedding dimension: (48, 96, 192, 384)
>
> > W7: The potential for leveraging the weights of larger models (which are more powerful) is high. However, Table 7 indicates that the proposed method struggles to effectively learn from these larger models.
>
> Weight selection is designed to find the best set of parameters in a larger model to initialize a small model. For ViT-S, approximately 1/4 of its parameters are utilized as initialization ViT-T. Comparatively, ViT-L has only 1/60 of its parameters utilized. Therefore, initializing ViT-T from ViT-L would naturally incur a higher information loss, which is the primary reason why it has a worse performance than ViT-S. This phenomenon can also be explained by the concept of information density. The knowledge acquired from the pretraining dataset is embedded in 20M parameters in ViT-S, but yet 300M parameters in ViT-L. Thus a parameter subset of 5M from ViT-S would carry more knowledge than the subset of the same size from ViT-L.
>
> In addition, weight selection can be **progressively applied to boost performance.** Directly initializing ViT-T from ViT-B reaches an accuracy of 77.6. However, if we initialize ViT-S from ViT-B, train ViT-S for 300 epochs, and then use the trained ViT-S to initialize ViT-T, ViT-T would reach a final accuracy of 80.4. With less information loss at each step, weight selection could help small models learn from larger models more effectively.
>
>
> | Setting                 | CIFAR-100 test acc |
> | ----------------------- |:------------------:|
> | random init             |        72.3        |
> | ViT-B -> ViT-T          |        77.6        |
> | ViT-B -> ViT-S -> ViT-T |        80.4        |
>
>
> > W8: Some grammar errors:
>
> Thanks, we have updated the paper to correct the grammar.
>
> We thank you again for your valuable feedback and we hope our response can address your questions. We have revised the paper accordingly and marked all changes in red. If you have any further questions or concerns, we are very happy to answer.

---

> > ### Comment · Reviewer_FWr5 · 2023-12-04
> >
> > Since the authors provide more empirical results, I would like to raise my score to 6. I hope the authors can revise this paper carefully with the new results.

---

### Official Review · Reviewer_HkYt · 2023-10-30

**Soundness:** 3 good
**Presentation:** 4 excellent
**Contribution:** 3 good
**Rating:** 8
**Confidence:** 4

**Summary:**

The paper introduces 'weight selection', a simple but effective method that uses the weight of pre-trained large model to initialize the small model. The authors have conducted extensive experiments, showcasing the method's superiority compared to initializing models from scratch.

**Strengths:**

1. The experiments are comprehensive, covering both vision transformers and CNNs. Additionally, they have delved into various aspects such as knowledge distillation, comparison with pruning methods, and linear probing, which offer valuable insights
2. The paper is easy to read and follow.

**Weaknesses:**

See questions.

**Questions:**

1. Could the authors give further explanation on the difference between 'consecutive selection' and 'uniform selection' in section 3.2?
2. For comparison with pruning, could the authors elaborate on how L1 pruning was implemented? I am asking the question since the embedding dimension is changed.

---

> ### Author Response · Authors · 2023-11-22
> **Rebuttal by Authors**
>
> We sincerely thank you for your constructive comments. We are encouraged that you find our paper presents comprehensive experiments and provides valuable insights from various aspects. We would like to address the comments and questions below:
>
> > Q1: Could the authors give further explanation on the difference between 'consecutive selection' and 'uniform selection' in section 3.2?
>
> Taking a 1-D vector [1, 2, 3, 4] as an example. If we would like to select a shorter vector of length 2 from this vector, consecutive selection means selecting consecutive indices like [1, 2], while uniform selection means selecting evenly spaced indices like [1, 3].
>
> For architectures with grouped components, like multi-head self-attention layers in ViT and grouped convolutional layers in ResNeXt, consecutive selection would mean selecting some complete groups while discarding other groups, while uniform selection will select a portion of all existing groups. In the context of weight selection from ViT-S to ViT-T, consecutive selection would approximately select 3 heads out of 6 heads in the multi-head self-attention layers, while uniform selection would select half of elements of each head. For architectures without grouped components, consecutive selection has no difference from uniform selection. As presented in Table 3, consecutive selection is of similar performance with uniform selection.
>
>
> > Q2: For comparison with pruning, could the authors elaborate on how L1 pruning was implemented? I am asking the question since the embedding dimension is changed.
>
> We perform L-1 pruning on the embedding dimension. Taking ViT-T as an example, we first perform L-1 pruning on the patch embedding layer on the embedding dimension. For the rest layers, it is followed by removing neurons on the input dimension corresponding to the pruned neurons of the previous layer, and then pruning the output dimension/channels based on $L_1$ norm. L-1 pruning would break the consistency on residual connections, and thus yield a worse result than uniform selection, consecutive selection, and random selection with consistency as presented in Table 3.
>
> We thank you again for your valuable feedback and we hope our response can address your questions. We have revised the paper and marked all changes in red. If you have any further questions or concerns, we are very happy to answer.

---

> > ### Comment · Reviewer_HkYt · 2023-11-22
> > **Thank you for your reply**
> >
> > After reading the additional experiments from the author and reviews from other reviewers, most of my concerns are addressed and I tend to keep my rating.

---

### Official Review · Reviewer_rXy8 · 2023-10-31

**Soundness:** 3 good
**Presentation:** 2 fair
**Contribution:** 3 good
**Rating:** 6
**Confidence:** 3

**Summary:**

This work introduces a method aimed at accelerating the convergence of training by transferring pre-trained knowledge from a different network. The approach involves extracting well-trained weights, which exhibit characteristics similar to high-pass filtering (HPF) or low-pass filtering (LPF), from a pretrained network and transplanting these weights onto a novel network, as opposed to starting with random initialization. This technique facilitates quicker training of the new architecture. The authors conduct comparisons between this method and random initialization, demonstrating its superior performance.


However, I have some concerns as follows:

1)	Firstly, it's important to note that the comparisons made in this work may not be entirely adequate. From my understanding, this approach falls somewhere between traditional initialization and transfer learning. Given that the method involves adopting a set of pre-trained filters or weights, it can be argued that this work is closer in nature to transfer learning rather than a simple initialization process. Therefore, it might be more appropriate to compare it with transfer learning methods rather than just random initialization. If the proposed method can approach or match the convergence levels of ImageNet pre-training, it would further support its practicality and utility in various applications, reinforcing its value in the field of deep learning.

2)	Comparing the proposed method directly to random initialization may not provide a fair assessment, given the distinct differences in the initialization processes. Unlike the proposed method, a network initialized randomly must construct its structural filters (HPF or LPF) from scratch, which typically requires some epochs and computational resources. In contrast, the proposed method can bypass this initial construction step and is expected to be faster.

3)	I strongly recommend reconsidering the categorization of this work from merely weight initialization to a more encompassing concept of knowledge transfer to a novel architecture. This work goes beyond traditional weight initialization methods by borrowing well-structured filters from a pre-trained network with a different architecture. This transfer of knowledge to a distinctly different architecture represents a significant departure from conventional weight initialization and indeed underscores the novelty of the approach. Categorizing it as a knowledge transfer method would better capture its unique contribution in the field.

4)	Building upon the suggested change in categorization to knowledge transfer, it raises an interesting question about the potential applicability of the proposed method to highly dissimilar architectures. For instance, exploring the feasibility of transferring knowledge from a pre-trained ResNet to a Vision Transformer (ViT) is a compelling idea like Model Evolution. Training ViT models often necessitates access to large-scale datasets, but they can offer superior performance in various tasks. If the proposed method could indeed facilitate the transfer of knowledge from a ResNet to a ViT, it would hold significant promise. This potential application could lead to advancements in leveraging the strengths of different architectures and addressing challenges related to dataset size, thus expanding the scope and impact of the proposed method.

I appreciate your insights, and it's clear that you find the work promising. However, I think that the manuscript doesn’t well categorize the proposed method and the experiments are not adequately conducted. I believe this work can be much improved.

**Strengths:**

See above

**Weaknesses:**

See above

**Questions:**

See above

---

> ### Author Response · Authors · 2023-11-22
> **Rebuttal by Authors**
>
> We sincerely thank you for your constructive comments. We are encouraged that you find our paper demonstrates unique insight. We would like to address the comments and questions below:
>
> > W1: Given that the method involves adopting a set of pre-trained filters or weights, it can be argued that this work is closer in nature to transfer learning rather than a simple initialization process. Therefore, it might be more appropriate to compare it with transfer learning methods rather than just random initialization.
>
> Weight selection comes into place when there are **no existing pretrained models** that fulfill specific hardware requirements (not small enough). If suitable pretrained models are available, their use is undoubtedly preferred. However, this is usually not the case: for instance, in the context of ViTs, the original ViT paper only provides ViT-B (80M) as its smallest model, as is the case with CLIP [Radford2021] or MAE [He2022] pretrained models. Similarly, in the context of language models, the smallest LlaMa model has 3 billion parameters, posing a challenge for most individual users. In our paper, we demonstrate the application of weight selection using ViT-T and ConvNeXt-F as examples, representing smaller models that were subsequently developed by the community following the initial proposal of the architecture. In practical settings, our method can be employed to construct smaller ViTs, ConvNets, LlaMas, and all other existing nerual network architectures.
>
> We conduct an ablation study on comparing weight selection to transfer learning pipeline (pretraining + finetuning). The experiment involved training the model on ImageNet for a specified number of epochs, followed by 300 epochs of fine-tuning on CIFAR-100. Given the significant difference in the number of training images between ImageNet (1.3 million) and CIFAR-100 (50,000), one epoch of training on ImageNet is approximately equivalent to 25.6 epochs of training on CIFAR-100.
>
>
> | Pre-train epochs  |  0   |  10   |  20   |  40  |  60  |  80  | 300  |
> | ----------------- |:----:|:-----:|:-----:|:----:|:----:|:----:|:----:|
> | CIFAR100 test acc | 72.4 | 76.3 | 77.2 | 80.0 | 81.5 | 82.5 | 85.7 |
>
> After 60 epochs of pretraining on ImageNet, the CIFAR-100 accuracy of ViT-T reaches 81.4, matching the accuracy achieved through weight selection. Notably, the accuracy obtained after 300 epochs of pretraining is equivalent to that achieved through fine-tuning. This demonstrates that weight selection effectively bridges the significant gap in accuracy observed during pretraining without incurring additional computational costs. **To achieve a comparable level of accuracy to weight selection through pretraining / transfer learning, one would need approximately 6.12 times more training resources and the access to the large pretraining dataset**, which could be unavailable like JFT-300M. Under the circumstance of no existing pretrained models of the desired size available, weight selection is far more resource-saving than transfer learning, and avoids the need for a large-scale pretraining dataset. We have included this result in section 5.
>
>
> > W2: Comparing the proposed method directly to random initialization may not provide a fair assessment, given the distinct differences in the initialization processes. Unlike the proposed method, a network initialized randomly must construct its structural filters (HPF or LPF) from scratch, which typically requires some epochs and computational resources. In contrast, the proposed method can bypass this initial construction step and is expected to be faster.
>
> Yes, we agree that the superiority of weight selection as initialization comes from the pretrained parameters/filters, and is thus expected to converge faster. In addition, if the initialization filters are obtained from a model trained on a larger dataset, weight selection will bring addtional benefit.
>
> | Model |  Teacher (weight selection)   | CIFAR-100 test acc |
> |:-----:|:-----------------------------:|:-----------------:|
> | ViT-T | ViT-S pretrained on ImageNet  |       81.4        |
> | ViT-T | ViT-S pretrained on CIFAR-100 |       77.6        |
>
> Our contribution mainly lies in practice. We identify the general need for smaller models, and propose weight selection as a cost-free add-on technique to develop smaller models with their existing larger versions available. We find weight selection extremely effective, capable of substituting the current common practice which uses random initialization when training smaller models.
>
> [Radford2021] Radford, Alec, et al. "Learning transferable visual models from natural language supervision." International conference on machine learning. PMLR, 2021.
>
> [He2022] He, Kaiming, et al. "Masked autoencoders are scalable vision learners." Proceedings of the IEEE/CVF conference on computer vision and pattern recognition. 2022.

---

> ### Author Response · Authors · 2023-11-22
> **Rebuttal by Authors**
>
> > W3: I strongly recommend reconsidering the categorization of this work from merely weight initialization to a more encompassing concept of knowledge transfer to a novel architecture. This work goes beyond traditional weight initialization methods by borrowing well-structured filters from a pre-trained network with a different architecture. This transfer of knowledge to a distinctly different architecture represents a significant departure from conventional weight initialization and indeed underscores the novelty of the approach. Categorizing it as a knowledge transfer method would better capture its unique contribution in the field.
>
> Thank you for your suggestion. **We would like to consider framing our method in a knowledge transfer context.** We also want to emphasize that in practice, our method serves as a cost-free initialization technique. Our method's benefit indeed comes from the knowledge transfer from large models to small models via parameters. In light of this, we are considering changing our title to **"initializing small models with larger ones"**, which captures the knowledge transfer from large models to small models as wellas showing it serves as an initialization technique.
>
> > W4: Building upon the suggested change in categorization to knowledge transfer, it raises an interesting question about the potential applicability of the proposed method to highly dissimilar architectures. For instance, exploring the feasibility of transferring knowledge from a pre-trained ResNet to a Vision Transformer (ViT) is a compelling idea like Model Evolution. Training ViT models often necessitates access to large-scale datasets, but they can offer superior performance in various tasks. If the proposed method could indeed facilitate the transfer of knowledge from a ResNet to a ViT, it would hold significant promise. This potential application could lead to advancements in leveraging the strengths of different architectures and addressing challenges related to dataset size, thus expanding the scope and impact of the proposed method.
>
> **We have conducted experiments on transferring knowledge from ConvNeXt to ViT via weight selection and obtained positive results.** More specifcally, we use the **pretrained isotropic ConvNeXt-S (20M parameters) to initialize ViT-T (5M parameters)** using weight selection. As ConvNeXt has identical architectures with ViT other than the self-attention layers, we are able to directly create a mapping between components.
>
> We initialize the patch embedding module, and all the fully connected layers of ViT-T with the corresponding component in the isotropic ConvNeXt-S.
>
> |               Setting               | CIFAR-100 test acc |
> |-----------------------------------|:-----------------------:|
> |             random init             |          72.3           |
> | cross-architecture weight selection |          74.8           |
>
> In this experiment, we did not initialize the self-attention layers of ViT-T with isotropic ConvNeXt-S, since there is no explicit way to initialize the self attention layers with convolutional layers as their weights and operations are completely different. Even without initializing self-attention layers, we observe a substantial improvement in self-attention layers. We hope this initial experiment can shed a light on transferring knowledge from ConvNets to vision transformers. Initializing self-attention layers with convolutional layers would be a part of our future works extending beyond weight selection.
>
> We thank you again for your valuable feedback and we hope our response can address your questions. We have revised the paper accordingly and marked all changes in red. If you have any further questions or concerns, we are very happy to answer.

---

> > ### Comment · Reviewer_rXy8 · 2023-11-23
> >
> > I appreciate the response.
> >
> > The latest version of the paper shows many improvements compared to the previous one, and it addresses several of my concerns. In particular, I find the feasibility of transferring from ConvNeXt to ViT quite intriguing.
> >
> > Given the thorough rebuttal provided and the authors' willingness to reconsider the categorization of their proposed method, I have decided to raise my score.
> >
> > Nevertheless, I still recommend that the authors conduct a comparative analysis with existing transfer learning methods, learnable initialization techniques, and distillation works as mentioned in the other reviewers' comments. For learning-based initialization, I recommend the following papers:
> >
> > 1) [Zhu 2021]. Gradinit: Learning to initialize neural networks for stable and efficient training. NeurIPS 2021
> > 2) [Dauphin 2019]. Metainit: Initializing learning by learning to initialize. NeurIPS, 32, 2019. 2019
> > 3) [Knyazev 2021]. A. Parameter prediction for unseen deep architectures. NeurIPS, 34:29433–29448, 2021.
> >
> > This comparison would further strengthen the paper by positioning the proposed method within the broader context of current research in the field.

---

### Official Review · Reviewer_A6vA · 2023-11-02

**Soundness:** 2 fair
**Presentation:** 4 excellent
**Contribution:** 3 good
**Rating:** 8
**Confidence:** 4

**Summary:**

The paper introduces a novel approach to initialize smaller models given large pretrained models. The approach is simple, efficient and shows good results compared to random init and other numerous baselines and many image classification tasks.

**Strengths:**

1. The problem of obtaining strong weights for small models given large pretrained models is important and solving it efficiently is challenging.

2. The paper explores a very interesting idea of simple weight copying from the source to target network as an efficient and training-free way to initialize smaller model.

3. The uniform selection approach is reasonable and takes into account multi-head structure and group convolution of source networks.

4. The paper has many nicely outlined and run experiments that overall look interesting and convincing enough.

5. The approach outperforms random init and works well on different vision tasks.

6. The paper is well presented and enjoyable to read.

**Weaknesses:**

1. The paper misses some important related works:
- [Czyzewski2022] Breaking the Architecture Barrier: A Method for Efficient Knowledge Transfer Across Networks
- [Chen2021] bert2BERT: Towards Reusable Pretrained Language Models
- [Chen2015] Net2net: Accelerating learning via knowledge transfer

Specifically, [Czyzewski2022] introduced a very similar weight transfer technique that seems more general than in the submitted paper since it allows for the source and target architectures to be different. They copied the center slices from source weights, which could be an interesting ablation in this submission. [Chen2015] proposed a simple approach to replicate weights to initialize models. While [Chen2015,Chen2021] focus on initializing a larger model, the idea of copying the weights from the source network to a target one is very similar and should be discussed in detail. This makes the novelty of the paper limited.

2. The paper also should report results of student models pretrained and transferred to smaller datasets. While this would not be a fair baseline to the proposed method since it requires expensive pretraining, it could give important clues about what is the best performance of the model on this task. In addition, some papers (e.g. [Knyazev2021]) reported that very little pretraining is required to achieve competitive transfer learning performance. Therefore, it's not clear if the proposed weight selection approach would be competitive if someone has resources to pretrain a small target net on some large data first (at least for a little bit).

- [Knyazev2021] Parameter Prediction for Unseen Deep Architectures

3. For structured pruning, there are many stronger methods than L1 pruning, e.g. [Kwon2022, Halabi2022] and the baselines therein. The results in Table 6 are quite close and L1 pruning is also very simple, therefore stronger pruning methods could potentially match the performance.

- [Kwon2022] A Fast Post-Training Pruning Framework for Transformers
- [Halabi2022] Data-Efficient Structured Pruning via Submodular Optimization

4. Most of the experiments are performed on transferring weights from ViT-S to ViT-T which have the same number of layers (12). It would be more interesting to see results when the source model is also deeper (which is usually the case in practice). Related to that, the teacher size results show that ViT-S is the best teacher and a potential reason for that could be that the layer selection approach (first-N) is not the best (perhaps a uniform selection of layers similarly how it is done for weights could be better than first N?). In Table 7, it would be very interesting to see the respective results for distillation which usually works better if the source model is larger/stronger (see the DeiT paper).

Despite all the weaknesses, I'm looking forward to the the authors' response and will be willing to update the rating.

**Questions:**

1. In Net2net [Chen2015], for initialization the copied weights are rescaled to take into account the width difference (i.e. for wider/thinner networks one needs smaller/larger weight values according to standard weight initialization such as K.He or X.Glorot). Have the authors tried to rescale copied weight values?

2. For "distill + weight selection" is the target model first initialized with weight selection and then the distillation is run or it's different?

3. It seems that the results of baselines (random init and mimetic init) are different from previous papers like the mimetic init paper even though the setup looks the same. For example, in the mimetic init paper for CIFAR-10 ViT-T the accuracy is 87.39 for rand init and 91.38 for mimetic init, while in this submission it is 92.4 and 93.3 respectively. Can the authors elaborate on the reasons for that?

4. In L1 pruning experiments, is the final number of selected weights the same as in the proposed weight selection? It's a little bit surprising to see that L1 underperforms uniform weight selection implying that the latter is actually a stronger pruning method. Can the authors elaborate on that?

---

> ### Author Response · Authors · 2023-11-22
> **Rebuttal by Authors**
>
> We sincerely thank you for your constructive comments. We are encouraged that you find our paper presents meaningful key ideas, provides abundant experimental results, applies interesting and reasonable selection methods, and is well-organized and nice to read. We would like to address the comments and questions below:
>
> > w1: This paper misses some important related works. Specifically, [Czyzewski2022] introduced a very similar weight transfer technique that seems more general than in the submitted paper since it allows for the source and target architectures to be different. They copied the center slices from source weights, which could be an interesting ablation in this submission. [Chen2015] proposed a simple approach to replicate weights to initialize models. While [Chen2015,Chen2021] focus on initializing a larger model, the idea of copying the weights from the source network to a target one is very similar and should be discussed in detail. This makes the novelty of the paper limited.
>
> Thanks for pointing out these related works. We have added the discussion to discuss these works to *Utilizing pretrained models* paragraph of the related work section.
>
> Weight selection, allows for the transfer of weights from larger models to smaller models applicable to **all kinds of architectures**, as long as the large model and small model belong to the same model family, while [Czyzewski2022] introduces parameter transfer across different convolutional neural networks. The center slices selection in this work is ablated as consecutive selection presented in Table 3.
>
> Our method aims to resolve the scenario where **a smaller version of an existing model** is needed whereas [Czyzewski2022]'s purpose is to transfer parameters to a **new convolutional architecture**. In practice, there is a relatively lower demand for developing entirely new models. We anticipate weight selection to be a practical and useful technique for developing smaller models of existing architectures.
>
> Our work differs from [chen2015] and [chen2021] in multiple aspects: targets, motivations for transferring knowledge, and the approach.
> - Our work focuses on efficiently developing **smaller models**, while these two works aim to accelerate training for **larger models**.
> - Weight selection draws knowledge from both larger models and their pretraining datasets, enabling smaller models to both converge faster and **achieve better performance**. In contrast, [chen2015] and [chen2021] primarily focus on **training efficiency** (convergence speed).
> - The weight selection process involves **selecting** an appropriate subset of pretrained parameters, while [chen2015] and [chen2021] transform smaller weight matrices into larger matrices in a larger network.
>
> [Czyzewski2022] Czyzewski, Maciej A., Daniel Nowak, and Kamil Piechowiak. "Breaking the Architecture Barrier: A Method for Efficient Knowledge Transfer Across Networks." arXiv preprint arXiv:2212.13970 (2022).
>
> [Chen2021] Chen, Cheng, et al. "bert2bert: Towards reusable pretrained language models." arXiv preprint arXiv:2110.07143 (2021).
>
> [Chen2015] Chen, Tianqi, Ian Goodfellow, and Jonathon Shlens. "Net2net: Accelerating learning via knowledge transfer." ICLR 2016.

---

> > ### Comment · Reviewer_A6vA · 2023-11-22
> >
> > I would like to thank the authors for an extensive response that addressed many of my concerns.
> >
> > One of the main concerns for me remains related work, specifically [Czyzewski2022]. Even though their method was only tested on convnets, it seems to be a general method that could be in principle applied to different architectures including scenarios of ViT-S -> ViT-T. Moreover, while "there is a relatively lower demand for developing entirely new models" is true, having such an ability might still be advantageous. So given small method differences compared to [Czyzewski2022], the novelty of this submission is limited. Nevertheless, this submission is still complementary to [Czyzewski2022] with extensive experiments and analysis showing interesting results and making useful contribution. Moreover, [Czyzewski2022] does not appear to be a published peer-reviewed paper so that some overlap might be fine.
> >
> > The authors response is long and requires more time to read it carefully, so I will consider revising the final rating once I read it in detail.

---

> > > ### Author Response · Authors · 2023-11-22
> > > **Further clarification on related works**
> > >
> > > We would like to thank you for your prompt response and your time into reading our extensive rebuttal. We are encouraged to see that you find our work provide interesting results and make useful contribution. We agree with you that, while differing in motivation and use cases, the proposed approach in [Czyzewski2022] can be applied to initialize ViT-T with ViT-S, and exhibits a similar selection criterion to the consecutive element selection approach in weight selection. However, we would like to clarify that our approach does not involve constructing a model zoo as needed in [Czyzewski2022] to initialize different modules, which can be very difficult to maintain as a large number of new models are developed.  We consider weight selection as a practical technique in this large-model era.
> > >
> > > We summarize our unique contribution as follows:
> > >
> > > - Identify the common scenario where smaller versions of existing architectures are in need of development.
> > > - Propose a practical initialization technique for this scenario without incurring any additional computation costs.
> > > - Identify the importance of maintaining consistency in element selection, and then standardize and simplify our method to uniform selection, which can be easily adopted.
> > > - Prove our approach's compatibility with knowledge distillation.

---

> ### Author Response · Authors · 2023-11-22
> **Rebuttal by Authors**
>
> > w2: the paper also should report results of student models pretrained and transferred to smaller datasets. While this would not be a fair baseline to the proposed method since it requires expensive pretraining, it could give important clues about what is the best performance of the model on this task. In addition, some papers (e.g. [Knyazev2021]) reported that very little pretraining is required to achieve competitive transfer learning performance. Therefore, it's not clear if the proposed weight selection approach would be competitive if someone has resources to pretrain a small target net on some large data first (at least for a little bit).
>
> Weight selection comes into place when there are **no existing pretrained models** that fulfill specific hardware requirements (not small enough). If suitable pretrained models are available, their use is undoubtedly preferred. However, this is usually not the case: for instance, in the context of ViTs, the original ViT paper only provides ViT-B (80M) as its smallest model, as is the case with CLIP [Radford2021] or MAE [He2022] pretrained models. Similarly, in the context of language models, the smallest LlaMa model has 3 billion parameters, posing a challenge for most individual users. In our paper, we demonstrate the application of weight selection using ViT-T and ConvNeXt-F as examples, representing smaller models that were subsequently developed by the community following the initial proposal of the architecture. In practical settings, our method can be employed to construct smaller ViTs, ConvNets, LlaMas, and all other existing nerual network architectures.
>
> To investigate your mentioned problem of comparing with "has some resource for pretraining", we conduct an ablation study focused on this factor. The experiment involved training the model on ImageNet for a specified number of epochs, followed by 300 epochs of fine-tuning on CIFAR-100. Given the significant difference in the number of training images between ImageNet (1.3 million) and CIFAR-100 (50,000), one epoch of training on ImageNet is approximately equivalent to 25.6 epochs of training on CIFAR-100.
>
>
> | Pretraining epochs  |  0   |  10   |  20   |  40  |  60  |  80  | 300  |
> | ----------------- |:----:|:-----:|:-----:|:----:|:----:|:----:|:----:|
> | CIFAR100 test acc | 72.4 | 76.26 | 77.17 | 80.0 | 81.5 | 82.5 | 85.7 |
>
> After 60 epochs of pretraining on ImageNet, the CIFAR-100 accuracy of ViT-T reaches 81.4, matching the accuracy achieved through weight selection. Notably, the accuracy obtained after 300 epochs of pretraining is equivalent to that achieved through fine-tuning. This demonstrates that weight selection effectively bridges the significant gap in accuracy observed during pretraining without incurring additional computational costs. **To achieve a comparable level of accuracy to weight selection through pretraining / transfer learning, one would need approximately 6.12 times more training resources and access to the large pretraining dataset**, which could be unavailable like JFT-300M. Under the circumstance of no existing pretrained models of the desired size, weight selection is far more resource-saving than transfer learning, and avoids the need for a large-scale pretraining dataset. We have included this result in section 5.
>
> [Radford2021] Radford, Alec, et al. "Learning transferable visual models from natural language supervision." International conference on machine learning. PMLR, 2021.
>
> [He2022] He, Kaiming, et al. "Masked autoencoders are scalable vision learners." Proceedings of the IEEE/CVF conference on computer vision and pattern recognition. 2022.

---

> ### Author Response · Authors · 2023-11-22
> **Rebuttal by Authors**
>
> > w3: For structured pruning, there are many stronger methods than L1 pruning, e.g. [Kwon2022, Halabi2022] and the baselines therein. The results in Table 6 are quite close and L1 pruning is also very simple, therefore stronger pruning methods could potentially match the performance.
>
> To the best of our knowledge, there is no existing pruning framework that can transform a ViT-S model into a ViT-T model. **Existing pruning methods operate within a block**. [Halabi2022] attempt to preserve layerwise output after pruning using submodular optimization. [Kwon2022] calculate the optimal mask in order to discard heads and filters within each layer. No existing pruning approaches modify the embedding dimension, as this is considered a destructive operation that would significantly degrade model performance.
>
> In the context of transforming larger models into smaller models involving a change in embedding dimension, we identify **maintaining consistency** (selecting the same indices for all weight matrices) as the critical factor. Specifically, for all networks with residual connections, the layer can be represented mathematically as:
>
> $$ y = F(x) + x $$
>
> where $x, y \in \mathbb{R}^D$, and $D$ is the embedding dimension. This equation can also be expressed element-wise:
>
> $$ y_i = F(x)_i + x_i$$
>
> for all i from 0 to $D - 1$.
>
> Traditional pruning frameworks do not alter D, and consequently, they do not directly modify this set of equations. Under our setting, which reduces D to a smaller value, selecting the same set of indices on the input neurons and along F(⋅) (maintaining consistency) will preserve the addition operations on the subset of the original residual connection operations.
>
> **Empirical results show that maintaining consistency is the most important factor.** As demonstrated in Table 2, random selection with consistency outperforms random selection without consistency by a significant margin (an accuracy improvement of 4%). Moreover, random selection with consistency exhibits comparable performance to uniform/consecutive selection. Therefore, we determine that maintaining consistency is the factor that contributes most to the performance. To simplify and standardize weight selection in practice, we adopt uniform element selection (which inherently preserves consistency) as the default method for weight selection.
>
> [Kwon2022] Kwon, Woosuk, et al. "A fast post-training pruning framework for transformers." Advances in Neural Information Processing Systems 35 (2022): 24101-24116.
>
> [Halabi2022] El Halabi, Marwa, Suraj Srinivas, and Simon Lacoste-Julien. "Data-efficient structured pruning via submodular optimization." Advances in Neural Information Processing Systems 35 (2022): 36613-36626.

---

> ### Author Response · Authors · 2023-11-22
> **Rebuttal by Authors**
>
> > w4-(1): Most of the experiments are performed on transferring weights from ViT-S to ViT-T which have the same number of layers (12). It would be more interesting to see results when the source model is also deeper (which is usually the case in practice).
>
> **ConvNeXt-F in main results include layer selection process**. Our main results in Table 2 show considerable improvement on both ViT-T and ConvNeXt-F, initialized from ViT-S and ConvNeXt-T respectively. Note that ConvNeXt-F is of depth [2, 2, 6, 2] for its four stages, and its weight selection teacher ConvNeXt-T is of depth [3, 3, 9, 3] for its four stages, which would involve layer selection process.
>
> Our ablation on layer selection further confirms the superiority of first-N layer selection as shown in Table 5. We present Table 5 below for reference. As illustrated in section 5, to examine the effect of layer selection, we create Vit-A, a variant of ViT with 6 layers, with other model configurations identical to ViT-T. We present weight selection results (from ViT-S to ViT-A, from ConvNeXt-T to ConvNeXt-F) with different layer selection approaches.
>
> | Setting / Model                            | ViT-A | ConvNeXt-F |
> | ------------------------------------------ |:-----:|:----------:|
> | random init                                | 69.6  |    81.3    |
> | first-N layer selection                    | 77.6  |    84.4    |
> | maximally-spaced / uniform layer selection | 76.7  |    83.2    |
>
> **Finetuning results show first-N layer selection is preferred.** To rule out the effect of element selection, we conduct experiments to evaluate results of directly finetuning different subsets of 6 layers out of 12 layers of a pretrained ViT-T model. We present the results as follows:
>
> | layer selection            | CIFAR-100 test acc |
> | -------------------------- |:------------------:|
> | first-N                    |        81.6        |
> | mid-N                      |        68.3        |
> | last-N                     |        62.0        |
> | maximally-spaced / uniform |        76.3        |
>
> We observe a clear trend favoring shallow layers, with first-N > middle-N > last-N. And maximally spaced/uniform layer selection is slightly worse than first-N layer selection. Based on this empirical result, we find first-N layer selection a suitable method in weight selection. We have included this result in Appendix B.
>
>
> > w4-(2): Related to that, the teacher size results show that ViT-S is the best teacher and a potential reason for that could be that the layer selection approach (first-N) is not the best (perhaps a uniform selection of layers similarly how it is done for weights could be better than first N?).
>
> ViT-S and ViT-B both comprise 12 layers, while ViT-L has 24 layers. Since there is no layer selection operation because there are the same numbers of layers in ViT-T, ViT-S, and ViT-B, the result that ViT-S is a better teacher than ViT-B supports our conclusion that the smaller teacher is better. We attribute ViT-S's superior performance as a teacher to its similarity in size to its student, which has minimal information loss during weight selection.
>
> However, when examining the effects of layer selection methods on initializing ViT-T with ViT-L, we observe distinct outcomes for different layer selection approaches.
>
> | layer selection| CIFAR-100 test acc  |
> | ------- | ---- |
> | first-N  | 76.9 |
> | mid-N  | 75.9  |
> | last-N  |  77.1 |
> |   maximally-spaced / uniform     |  77.5   |
>
> We anticipate that **the best approach for layer selection might be related to the ratio between teacher and student**. We have added discussions for this phenomenon in appendix B. Our main approach would still adopt first-N layer selection as the default method, for the reason that in the case of a smaller teacher (which is favored in weight selection), first-N layer selection consistently outperforms its alternatives as shown in the answer to your previous question.

---

> ### Author Response · Authors · 2023-11-22
> **Rebuttal by Authors**
>
> > w4-3: In Table 7, it would be very interesting to see the respective results for distillation which usually works better if the source model is larger/stronger (see the DeiT paper).
>
> We present results for feature-based knowledge distillation on CIFAR-100 for ViT-S, ViT-B, and ViT-L.
>
> | Setting/model    | ViT-S | ViT-B | ViT-L |
> | --- | ----- | ----- | ----- |
> |  distill  |   78.4    |   76.3    |   76.8    |
> |  weight selection   |   81.4    |    77.6   |   76.9    |
> |   distill + weight selection  |   83.9    |    78.7   |    78.4   |
>
> Unlike logit-based non-transfer learning setting in DeiT, feature-based knowledge distillation has a similar preference on a smaller teacher with closer size to student. For example in [Ren2023], the best result for ViT-T is obtained by  distilling from ViT-S. Smaller teacher is favored by both weight selection and distillation. These results also show that weight selection's compatibility with knowledge distillation is consistent across teachers of different sizes.
>
> [Ren2023] Ren, Sucheng, et al. "TinyMIM: An empirical study of distilling MIM pre-trained models." Proceedings of the IEEE/CVF Conference on Computer Vision and Pattern Recognition. 2023.
>
> > Q1: In Net2net [Chen2015], for initialization the copied weights are rescaled to take into account the width difference (i.e. for wider/thinner networks one needs smaller/larger weight values according to standard weight initialization such as K.He or X.Glorot). Have the authors tried to rescale copied weight values?
>
> Yes, we have tried to rescale the copied weights based on kaiming initialization [He2015]. Specifically, if the desired student matrix size is of input dimension $s_{in}$, and teacher matrix is of input size $t_{in}$, we scale the result of weight selection by $\sqrt{t_{in}} / \sqrt{s_{in}}$. We report results of initializing ViT-T with different teachers with or without scaling.
>
> | Setting/teacher model                     | ViT-S | ViT-B | ViT-L |
> | --------------------------------- | ----- | ----- | ----- |
> | weight selection (no scaling)        | 81.4  | 77.6  | 76.9  |
> | weight selection (with scaling) | 81.8  | 75.8  | 76.9  |
>
> We find that this approach does not provide stable improvement, possibly due to the existence of normalization layers.
>
> [He2015] He, Kaiming, et al. "Delving deep into rectifiers: Surpassing human-level performance on imagenet classification." Proceedings of the IEEE international conference on computer vision. 2015.
>
> > Q2: For "distill + weight selection" is the target model first initialized with weight selection and then the distillation is run or it's different?
>
> Yes, your current understanding is correct. Other than the initialization of the student, the distillation trial is identical to the teacher. We have edited section 4.3 to clarify this.
>
> > Q3: It seems that the results of baselines (random init and mimetic init) are different from previous papers like the mimetic init paper even though the setup looks the same. For example, in the mimetic init paper for CIFAR-10 ViT-T the accuracy is 87.39 for rand init and 91.38 for mimetic init, while in this submission it is 92.4 and 93.3 respectively. Can the authors elaborate on the reasons for that?
>
> **Mimetic initialization has less gain under stronger training recipe.** [Asher2023] used 100 epochs for training, input size as 32, and RandAug and Cutout as data augmentation. Our training recipe is stronger: we use 300 epochs for training, 224x224 input size, and RandAug, Mixup, and Cutmix for data augmentation.  This conclusion is supported by [Asher2023]: *on ImageNet, using ResNet pipeline gives 4.1% improvement, while using DeiT pipeline (a stronger training recipe) gives 0.5% improvement.*
>
> [Asher2023] Trockman, Asher, and J. Zico Kolter. "Mimetic Initialization of Self-Attention Layers."

---

> ### Author Response · Authors · 2023-11-22
> **Rebuttal by Authors**
>
> > Q4: In L1 pruning experiments, is the final number of selected weights the same as in the proposed weight selection? It's a little bit surprising to see that L1 underperforms uniform weight selection implying that the latter is actually a stronger pruning method. Can the authors elaborate on that?
>
> In all existing pruning frameworks applicable to architectures with **residual connections**, they **do not prune the embedding dimension**, since that would incur a drastic performance degrade. Under the setting that embedding dimension needs to be pruned, we find consistency the most important factor, where the same indices should be pruned along the embedding dimension. Following L1 pruning approach would inevitably lose consistency, which is the primary reason causing the performance to drop. Empirical results in Table 3 show the importance of consistency.
>
> We thank you again for your valuable feedback and we hope our response can address your questions. We have revised the paper accordingly and marked all changes in red. If you have any further questions or concerns, we are very happy to answer.

---

### Author Response · Authors · 2023-11-22
**Invitation for Dicussion**

Dear Reviewers,

We would like to express our gratitude for your insightful feedback and suggestions, which have been very helpful in updating and enhancing our submission during the rebuttal process. We kindly invite you to review our author rebuttal so that we may address any further questions you may have or clarify any points that remain unclear. In summary, our rebuttal includes the following:

- We consider framing our method in knowledge transfer context in addition to an initialization method. We are considering changing our title to "initializing small models with larger ones". (R-rXy8)
- We conduct experiments to show the capability of weight selection applied to cross-architecture knowledge transfer. (R-rXy8)
- We present evaluation of weight selection for more architectures: ResNet, MLP-Mixer, Swin-Transformer, and Pyramid Vision Transformer. (R-FWr5)
- We add discussions for works that include copying parameters in the related work section. (R-A6vA)
- We conduct experiments to compare with transfer learning and show weight selection's effectiveness in saving training resources. (R-A6vA, R-rXy8)
- We emphasize and justify uniform selection from the perspective of maintaining consistency on residual connections. (R-A6vA, R-FWr5)
- We clarify how weight selection is combined with knowledge distillation. (R-A6vA)
- We conduct experiments for independent evaluation of layer selection, further justifying first-N layer selection approach. (R-A6vA, R-FWr5)
- We provide knowledge distillation results for larger teachers on ViT, and extend knowledge distillation results to ConvNeXt. (R-A6vA, R-FWr5)
- We show that weight selection can be applied progressively to better learn from larger models. (R-FWr5)
- We examine different layer selection approaches for larger teachers, and find that different student-teacher ratio result in different preference of layer selection approaches. (R-A6vA)
- We clarify the different results between mimetic initialization paper and our paper on CIFAR-10. (R-A6vA)
- We present weight selection results for teachers with different pretraining datasets. (R-rXy8)
- We present experiment results on rescaling parameters. (R-A6vA)
- We provide further explanation on the distinction between consecutive selection and uniform selection. (R-HkYt)
- We clarify our approach on applying L-1 pruning on the embedding dimension (R-HkYt)
- We add model configurations for architectures included in the main results. (R-FWr5)
- We clarify that smaller teacher is preferred in weight selection instead of larger teacher. (R-rXy8)
- We correct grammars. (R-FWr5)

We hope our responses can adequately address your concerns. We have integrated results in the rebuttal into the revision and marked in red. We sincerely appreciate your valuable feedback.

Best,

Authors

---

### Meta-Review · Area_Chair_Bh2P · 2023-12-10

**Metareview:**

All reviewers found this work solid and of broad interest. The claims are supported. The extensive rebuttal clarified the reviewers' concerns and the additional experimental results were appreciated. The latest version of the paper shows many improvements compared to the submitted one, resolving the open questions and providing convincing evidence of the effectiveness of the proposed approach. I would encourage the authors to incorporate the new results in the paper (or appendix) to further strengthen the paper.

**Justification For Why Not Higher Score:**

While interesting and solid, the paper required a relatively large number of clarifications and improvements. New experimental results had to be provided to support some of the claims and some related work was missing. At least one reviewer would like to have seen one additional set of comparisons.

**Justification For Why Not Lower Score:**

All reviewers agreed that post rebuttal this is a solid paper. The proposed method is of broad applicability and interest. A spotlight would increase visibility and awareness of this piece of work.

---

### Decision · Program_Chairs · 2024-01-16

Accept (spotlight)